# Niemann-Pick C-like Endolysosomal Dysfunction in DHDDS Patient Cells, a Congenital Disorder of Glycosylation, Can Be Treated with Miglustat

**DOI:** 10.3390/ijms26041471

**Published:** 2025-02-10

**Authors:** Hannah L. Best, Sophie R. Cook, Helen Waller-Evans, Emyr Lloyd-Evans

**Affiliations:** Medicines Discovery Institute, Main Building, Cardiff University, Cardiff CF10 3AT, UK; besth@cardiff.ac.uk (H.L.B.); cooks10@cardiff.ac.uk (S.R.C.); waller-evansh@cardiff.ac.uk (H.W.-E.)

**Keywords:** miglustat, lysosomal storage disease, DHDDS, Niemann-Pick, glycosphingolipids, cholesterol

## Abstract

DHDDS (dehydrodolichol diphosphate synthetase) and NgBR (Nogo-B Receptor) collectively form an enzymatic complex important for the synthesis of dolichol, a key component of protein N-glycosylation. Mutations in *DHDDS* and the gene encoding NgBR (*NUS1)* are associated with neurodevelopmental disorders that clinically present with epilepsy, motor impairments, and developmental delay. Previous work has demonstrated both DHDDS and NgBR can also interact with NPC2 (Niemann-Pick C (NPC) type 2), a protein which functions to traffic cholesterol out of the lysosome and, when mutated, can cause a lysosomal storage disorder (NPC disease) characterised by an accumulation of cholesterol and glycosphingolipids. Abnormal cholesterol accumulation has also been reported in cells from both individuals and animal models with mutations in NUS1, and suspected lipid storage has been shown in biopsies from individuals with mutations in DHDDS. Our findings provide further evidence for overlap between NPC2 and DHDDS disorders, showing that DHDDS patient fibroblasts have increased lysosomal volume, store cholesterol and ganglioside GM1, and have altered lysosomal Ca^2+^ homeostasis. Treatment of DHDDS cells, with the approved NPC small molecule therapy, miglustat, improves these disease-associated phenotypes, identifying a possible therapeutic option for DHDDS patients. These data suggest that treatment options currently approved for NPC disease may be translatable to DHDDS/NUS1 patients.

## 1. Introduction

De novo mutations in the genes’ *DHDDS* encoding dehydrodolichol diphosphate synthetase (DHDDS) and *NUS1* (nuclear undecaprenyl pyrophosphate synthase 1) encoding the Nogo-B receptor (NgBR) cause neurodevelopmental syndromes largely characterised by epilepsy and movement abnormalities including mild ataxia, myoclonus, and dystonia [1,2,3]. Although most cases of DHDDS and NUS1 syndrome are heterozygous, some instances of homozygosity have also been reported, whilst compound heterozygosity is rare and associated with a more severe disease presentation [4,5,6]. The DHDDS and NgBR protein products act together as subunits of the cis-prenyltransferase (cis-PTase) enzyme acting within the mevalonate pathway, a metabolic pathway responsible for the production of isoprenoids and sterols [7,8,9,10]. *DHDDS* encodes the catalytic subunit responsible for the synthesis of polyprenyl diphosphate, a precursor of dolichol monophosphate which is a lipid that is, in turn, important for protein N-glycosylation in the endoplasmic reticulum (ER) [11,12]. For these reasons, mutations in DHDDS and NUS1 are classed as congenital disorders of glycosylation (CDGs), a family of diseases caused by mutations in genes associated with N-glycan biosynthesis [13,14,15]. Surprisingly, whilst homozygosity for NUS1 or DHDDS is associated with phenotypes resembling a CDG [13,16], heterozygous DHDDS patients do not appear to present with any significant serum glycoprotein hypoglycosylation and, in contrast to other CDGs, urinary dolichol D18/D19 ratios and serum transferrin N-glycosylation profiles are normal, suggesting that other factors may contribute to the pathogenesis of this group of disorders [1,6,17].

It has been shown by several groups that both DHDDS and NgBR bind to the lysosomal cholesterol transport protein NPC2. This specific interaction between DHDDS and NPC2 was first identified from a yeast two-hybrid screen and confirmed by co-IP [18]. An interaction between NgBR and NPC2 was first shown in 2009 [19]. This later study also provided evidence that the NgBR interaction stabilised NPC2, as knock down of NgBR resulted in reduced NPC2 protein levels whilst increased expression of NgBR resulted in retention of NPC2 in the ER and prelysosomal compartments. Finally, the presence of free cholesterol accumulation, similar to that observed in cells from individuals with mutations in NPC2, was observed in a human umbilical vein endothelial cell (HUVEC) line following knock down of NgBR and in murine embryonic fibroblasts heterozygous for *NUS1* [19]. Furthermore, the accumulation of cholesterol has subsequently been reported in *NUS1* patient fibroblasts, *nus1* morphant zebrafish [20], and in the orthologous drosophila model [21], whilst altered lysosomes containing intramembraneous lipid whorls have been noted in DHDDS patient biopsies, suggestive of glycosphingolipid (GSL) storage [1].

GSL accumulation is a key hallmark of Niemann-Pick C (NPC) disease, an inherited lysosomal disease caused by mutation and loss of function of the lysosomal cholesterol and lipid handling NPC1 and NPC2 proteins [22]. Despite evidence that disruption of the functional interaction between the subunits of cis-PTase and NPC2 results in NPC2 mis-localisation to the ER, no study has yet confirmed the presence of other hallmarks of NPC disease cellular pathogenesis in DHDDS or NUS1 cells, namely lysosomal storage of other lipids including GSL and lyso-(bis)phosphatidic acid, endosomal transport-autophagy abnormalities, and lysosomal Ca^2+^ signalling defects [22,23,24]. Miglustat was the first approved disease-modifying therapeutic for NPC. It is a small molecule inhibitor of glucosylceramide synthase, the enzyme which catalyses the first step in the synthesis of glycosphingolipids through the addition of glucose to ceramide [25]. There are now multiple approved disease-modifying therapies for NPC disease, including arimoclomol [26] and N-acetyl-L-leucine [27]. Investigating the presence of NPC phenotypes in DHDDS cells, where NPC2 function may be impaired, is critical for determining whether these NPC therapies could be repurposed for DHDDS/NUS1 syndrome where there is currently no disease-modifying therapy.

## 2. Results

### 2.1. NPC2 Human Fibroblasts Present with Classical NPC Disease Cellular Phenotypes

Whilst phenotypic characterisation of NPC1 mutant human fibroblasts has been reported frequently in the literature [28,29,30,31], to the best of our knowledge, beyond analysis of the NPC2 hypomorph mouse model [32], no one has characterised the presence of phenotypes—other than cholesterol storage—in NPC2 patient fibroblasts [33]. Furthermore, no one, to our knowledge, has confirmed that miglustat mediates benefits in any model of NPC2 disease. To generate a point of reference for the phenotypic analysis of the DHDDS patient cells, we first determined what phenotypes were present in NPC2 disease. As expected, we observed a clear accumulation of cholesterol (∼12-fold) and ganglioside GM1 (∼3 fold) in punctate peri-nuclear lysosomal structures in NPC2 patient fibroblasts (Figure 1). This lipid accumulation was associated with elevated lysotracker fluorescence (∼2.6 fold) and an accumulation of autophagosomes (∼3.8-fold, Figure 1). Whilst these phenotypes are observed in many lysosomal diseases, albeit often at lower levels (e.g., cholesterol) [34], one phenotype that is relatively specific for NPC disease is the reduction of lysosomal Ca^2+^ content caused by the lysosomal accumulation of sphingosine [23]. Using GPN to perforate the lysosome in cells pre-clamped with ionomycin, we observed a clear reduction in lysosomal Ca^2+^ content in the NPC2 cells (Appendix A). In combination with significant intra-lysosomal accumulation of cholesterol and gangliosides, these phenotypes are classical hallmarks of NPC disease and are indeed present in both NPC1 and NPC2 patient cells, as would be expected. Miglustat is an approved therapeutic for NPC disease that works by reducing GSL biosynthesis via inhibition of glucosylceramide synthase. Treatment of NPC2 disease cells with miglustat, at 50 μM for 14 days, was sufficient to improve all the reported phenotypes except for cholesterol storage (Figure 1). This is expected, given that miglustat does not alter cholesterol accumulation in NPC1 mutant cells [25].

### 2.2. DHDDS Human Fibroblasts Cell Present with Lysosomal Lipid Storage

For comparative phenotyping assessment in DHDDS patient fibroblasts, we used four cell lines, which cover two different mutations (Table 1). Whilst cholesterol accumulation has been shown previously across several NUS1 models [20,21], it has not been fully investigated or confirmed as being lysosomal in DHDDS.

Combined filipin and LAMP1 immunofluorescence confirmed a ∼3.4-fold increase in lysosomal cholesterol in all four DHDDS patient cell lines (Figure 2A,B). Furthermore, LAMP1 was elevated ∼2.8 fold in all four cell lines, indicating an increased number/volume of lysosomes, presumably as a downstream effect of lipid storage (Figure 2A,C). Increased filipin staining colocalises strongly with LAMP1, indicating a lysosomal accumulation of cholesterol in DHDDS, as highlighted in Figure 2A,D. Miglustat treatment has no effect on cholesterol accumulation, which is unsurprising given that we know it has no beneficial effect in NPC disease models. LAMP1 also remained unchanged with miglustat treatment.

Using combined CtxB and LAMP1 staining, we went on to examine the intracellular distribution of a glycosphingolipid, ganglioside GM1 (Figure 3A). In the apparently healthy control cells, ganglioside GM1 (as indicated by FITC-conjugated cholera toxin) is present in cellular structures that do not predominantly colocalise with the lysosomal LAMP1 marker (see absence of yellow colour in Figure 3 GM1/LAMP1 merged panel). Ganglioside GM1 is likely present predominately in the secretory system where it is made [35,36]. In the DHDDS cell lines, we see an accumulation predominantly in the late endocytic system, indicated by an overlap of GM1 with perinuclear punctate LAMP1-positive structures (see increased yellow colour in GM1/LAMP1 merged panels, Figure 3A). Miglustat treatment significantly reduced the punctate staining of ganglioside GM1 by ∼1.5 fold, confirming that miglustat inhibits GSL synthesis, resulting in reduced lysosomal storage in DHDDS patient cells (Figure 3B). Whilst miglustat treatment decreased GSL storage, LAMP1 remained unchanged, as quantified in Figure 2, suggesting lysosomal expression of LAMP1 does not decrease in the same timeframe.

After confirmation of lipid storage, we next looked at LysoTracker, a marker of lysosomal volume that serves as a surrogate indicator for lysosomal storage and is commonly used as a biomarker for NPC and other lysosomal storage disorders [37] (Figure 4A). LysoTracker was significantly elevated by ∼1.7 fold in all DHDDS patient lines, confirming the increased lysosomal volume, as expected from previously observed lipid storage. Miglustat treatment significantly reduced LysoTracker area by ∼1.4 fold, indicating decreased lysosomal volume (Figure 4B). The reduction of LysoTracker in the absence of a LAMP1 reduction suggests lysosomes are reduced in size but are not being cleared within the 14-day treatment window.

### 2.3. DHDDS Cells Have Reduced Lysosomal Ca^2+^ Content

The storage of lipids, specifically sphingosine, in NPC causes a reduction in lysosomal Ca^2+^ content. To investigate lysosomal Ca^2+^ stores in DHDDS, cells were loaded with the cell permeable Ca^2+^ indicator, Fura2,AM. Cells were first treated with ionomycin to empty non-lysosomal intracellular Ca^2+^ stores, followed by GPN to perforate the lysosomal membrane and glycocalyx and release lysosomal Ca^2+^ into the cytoplasm (Figure 5A) [38,39]. DHDDS cells had approximately a 50% reduction in lysosomal Ca^2+^ content, which was partially corrected (approximately 30%) with miglustat (Figure 5B). Not only does this demonstrate another similarity to NPC but also indicates that miglustat could provide a functional benefit in DHDDS fibroblasts.

### 2.4. DHDDS Cells Have Altered Autophagic Flux

Finally, we looked at autophagy using the CYTO-ID autophagy probe, another marker which is frequently utilised in lysosomal storage disorders (Figure 6A). Autophagic vacuoles were increased by ∼2.8 fold across all four patient lines, indicating an upregulation of autophagy or an accumulation and failure to clear autophagic vacuoles. Whilst treatment with miglustat results in a qualitative visible reduction in CYTO-ID intensity between DHDDS cells and apparently healthy controls, there is no significant reduction in the spot area (Figure 6B). Further analysis did indicate a statistically significant decrease in spot intensity in three patient lines. Further work is needed to determine whether the flux of autophagy is impaired in DHDDS and how this might have an impact on miglustat treatment.

## 3. Discussion

In this report, we have demonstrated for the first time, to our knowledge, the presence of a plethora of NPC disease-like phenotypes in DHDDS patient cells. Albeit milder than those we observe in NPC2 loss-of-function patient cells, all four DHDDS cell lines presented with lysosomal expansion, storage of cholesterol and ganglioside GM1, and increased autophagic vacuoles. Furthermore, we observe a defect in the maintenance of lysosomal Ca^2+^ homeostasis, demonstrating a pathogenic change in cellular signalling, which we know from previous work is a direct consequence of sphingosine storage [23]. Our observations are based on data from fibroblast cells from four patients harbouring two different mutations, located in one of each of the two substrate binding sites present in the cis-PTase active site. The functional defect from these mutations is clear, due to both R37 and R205 participating in substrate binding [10]. Both mutations have been reported to result in a substantial decrease in cis-PTase activity to near the lower end of the detectable range (Table 1) [1,10]. Phenotypically, this study shows that both mutations are broadly similar, which is perhaps expected given that both mutations result in a similar functional perturbation.

Whilst these phenotypes are not entirely unique to NPC, as lipid storage, lysosomal expansion and autophagic vacuole accumulation are reminiscent of most lysosomal storage disorders (LSD), the additional presence of reduced lysosomal Ca^2+^ is a phenotype that has been reported in very few LSDs. Altered lysosomal Ca^2+^ homeostasis was first associated with loss-of-function of NPC1 [23] and, as we now show here, NPC2. The loss of NgBR function has previously been linked to the destabilisation and reduced expression of NPC2, leading to cholesterol storage [19]. The mechanism by which NPC2 may have reduced function in DHDDS is yet to be elucidated. Previous studies have demonstrated that an over-expression of NgBR results in increased ER localisation and stabilisation of NPC2 [19]. It is therefore tempting to speculate that levels of NgBR or NgBR-NPC2 interaction may be elevated in DHDDS as a compensatory mechanism, thus impacting NPC2 localisation. However, previous work has also shown that the overexpression of DHDDS or NgBR results in enhanced levels of its binding partner, indicating they act to stabilise each other on the formation of the cis-PTase enzyme complex, so a reduced expression of DHDDS can decrease NgBR expression [40]. Further highlighting the link between the disorders, NPC1 mutant mice show a modified pattern of mevalonate pathway lipids and decreased cis-PTase activity [41], an overall increase in dolichyl phosphate due to a decreased rate of degradation, and a reduction in lysosomal dolichol [41,42]. NPC2 disease has not been looked at in terms of mevalonate pathway lipids, although excess dolichol does not appear to impact NPC2-dependent cholesterol transfer between phospholipid vesicles [43]. Understanding how the expression and localisation of NgBR and, in turn, NPC2 are affected by DHDDS mutations is an interesting avenue for future investigational and therapeutic studies.

Whilst an accumulation of cholesterol has been previously noted in NUS1 patient fibroblasts, nus1 knock-down zebrafish [20], and the NUS1-orthologue fruit fly model [21], it has not been reported to the same extent for DHDDS. The work presented here supports a previous study that identified enlarged lysosomes containing electron-dense material as well as deposits of membranous whorls in a patient skin biopsy carrying an R211Q mutation [1]. Membranous whorls and electron-dense material are commonplace in lysosomal storage patients [44,45], with the whorl-like structures being attributed to gangliosides [46]—such as the ganglioside GM1 we identified as elevated in DHDDS-patient lysosomes. We identified one previous report indicating the presence of elevated filipin staining in DHDDS fibroblasts from a heterozygous-affected individual harbouring the R205Q mutation, and a minimal elevation in fibroblasts from a patient harbouring a heterozygous D95N mutation [3]. Our work corroborates the significantly elevated filipin in patients harbouring a R205Q mutation and now confirms it is lysosomal. The comparatively minor filipin elevation present in the D95N is associated with a milder disease course, as suggested by the later onset of tonic-clonic seizures and ataxia. This correlation suggests lysosomal storage could be a biomarker of DHDDS, although this would need substantially more investigation.

The identification of NPC disease-like phenotypes, alongside the established interaction between NPC2 and DHDDS proteins [18], raises the possibility that therapeutics approved for NPC could also be beneficial to DHDDS and indeed NUS1. Here, we show that miglustat, the first approved therapeutic for NPC, is effective at reducing glycolipid storage and lysosomal swelling in DHDDS. Furthermore, it partially rescues the lysosomal Ca^2+^ phenotype, demonstrating that miglustat may provide a functional improvement. In DHDDS, miglustat does not affect cholesterol storage, which is consistent with its limitations in NPC disease [24,25]. Whilst future studies will be required to assess the long-term effect of miglustat in DHDDS, more than 20 years of NPC patient-recorded use of miglustat indicates that reducing sphingolipid storage is beneficial and that chronic miglustat use is safe. In NPC patients, miglustat has been shown to stabilise disease progression and even provide improvement, especially in those with milder disease progression where an increase in lifespan has now been reported [47,48]. Very recently, two other small molecules were also approved for the treatment of NPC: arimoclomol [26] and N-acetyl-L-leucine [27]. Whilst the mechanism of action for both compounds is unclear, they are both reported to address, to some degree, the pathogenic lipid storage. Indeed, N-acetyl-L-leucine is also reported to slightly reduce cholesterol levels [49]. Both these compounds may be of interest for future DHDDS/NUS1 therapeutic studies.

## 4. Materials and Methods

### 4.1. Cell Maintenance and Drug Treatment

Apparently healthy (control) human (GM05399) and NPC2 patient (GM18455) fibroblast cell lines were obtained from the Coriell Cell Repository (Camden, NJ, USA); the DHDDS patient fibroblasts (Table 1) were from Dr. Frances Elmslie (St. George’s University Hospital, London, UK) and Dr. Eva Morava (Icahn School of Medicine at Mount Sinai). The fibroblasts were grown as monolayers in Dulbecco’s modified Eagle’s high glucose medium (DMEM; Life Technologies, Carisbad, CA, USA), supplemented with 10% foetal bovine serum (FBS; PAN biotech, Aidenbach, Germany) and 1% L-glutamine (complete medium). The cultures were maintained in a humidified 5% CO_2_ incubator at 37 °C. Miglustat (Toronto Research Chemicals, North York, ON, Canada, M344225) was dissolved in mqH_2_O to generate a 100 mM stock solution, which was stored at –20 °C. For miglustat treatment, the stock was diluted in complete media to a final concentration of 50 μM. Fresh miglustat-containing media was added every third day for 14 days prior to live staining, fixation, and imaging. For all experiments, the fibroblasts were passage matched with equal cell numbers seeded, and the cells were plated on Ibidi chamberslides for Ca^2+^ imaging or Perkin Elmer 96-well PhenoPlates for fluorescent stain imaging.

### 4.2. Determination of Lysosomal Ca^2+^

To determine lysosomal Ca^2+^ levels, cells were first loaded with the membrane permeant ratiometric Ca^2+^-binding dye Fura-2,AM (ab120873, Abcam, Cambridge, UK). The cells were washed and incubated for 45 min at room temperature in DMEM with 10% FBS, 1% BSA (Sigma-Aldrich, Gillingham, UK), 0.025% Pluronic F127 (Sigma-Aldrich, Gillingham, UK), and 5 mM Fura-2,AM. Following incubation, the cells were washed once and incubated for a further 10 min at room temperature in DMEM with 10% FBS to allow de-esterification of the dye. The cells were then washed and imaged in Hank’s balanced salt solution supplemented with 5 mM HEPES (pH7.4), 1 mM MgCl_2_ and 1 mM CaCl_2_. Imaging was performed in low Ca^2+^ buffer (5 mM HEPES (pH7.4), 1 mM MgCl_2_ and 5 μM CaCl_2_) on a Zeiss Axio Observer A1 microscope with Colibri LED illumination and high speed Axiocam Mrm camera. Fura-2,AM was excited at 360 and 380 nm LED wavelengths with emission monitored at 510 nm. Regions of interest were placed over whole cells, and videos were recorded using Axiovision 4.8.2 imaging software at intervals of 1–2 s following additions. To release lysosomal Ca^2+^, the cells were first incubated with the Ca^2+^ ionophore ionomycin (I9657, Sigma-Aldrich, Gillingham, UK), which does not impact lysosomes but causes other membranes to be permeant to Ca^2+^, followed by the cathepsin C substrate Gly-Phe-β-naphthylamide (GPN; ab145914, Abcam, Cambridge, UK), whose products induce perforation of the lysosomal membrane, triggering the release of the intralumenal Ca^2+^. The data were analysed in Microsoft Excel and Graphpad Prism (v10.1.0); Ca^2+^ traces are shown as the ratio of emission at 360/380 nm.

### 4.3. Lysosomal Staining with Lysotracker Red

Lysosomal density and distribution were determined with live imaging using the lysosomal dye LysoTracker Red DND-99 (L7528, Life Technologies, Inchinnan, UK). The cells were loaded at room temperature with 200 nM Lysotracker red in complete media for 10 min. Following loading, the cells were washed once in DPBS, counterstained with the nuclear marker Hoechst 33341 (0.4 mg/mL in DPBS; Life Technologies, Inchinnan, UK) for 10 min at room temperature, washed a further two times in DPBS, and imaged live using an Operetta high-content imaging system (Revvity, Llantrisant, UK).

### 4.4. Autophagic Vacuole Staining with CytoID

The cellular content of autophagic vacuoles, under rest, was determined using the fluorescent small molecule marker CYTO-ID from the CYTO-ID detection kit (ENZ-51031, ENZO life sciences, Exeter, UK) in live cells. The cells were incubated with CYTO-ID according to the manufacturer’s instructions. Briefly, CYTO-ID was diluted 1:1000 in a complete medium, and the cells were incubated for 30 min at room temperature prior to washing in DPBS and counterstaining with Hoechst, as described above. The cells were imaged live using the Operetta high-content imaging system (Revvity, Llantrisant, UK).

### 4.5. Staining of Cellular Lipid Content

To determine lipid content and localisation, we first washed cells in DPBS followed by fixation in 4% paraformaldehyde for 10 min at room temperature. The cells were subsequently washed twice with DPBS. To image cholesterol, the cells were stained with the autofluorescent polyene antibiotic filipin complex (BF162725, Biosynth, Compton, UK) at a concentration of 187.5 mg/mL in DMEM with 10% FBS for 30 min in the dark at room temperature. The cells were then washed once with DMEM with 10% FBS and twice with DPBS. Prior to imaging, the cells were counterstained with DRAQ5 nuclear marker (ab108410, Abcam, Cambridge, UK). To image glycosphingolipids, the cells were stained with FITC labelled Cholera toxin subunit B (CtxB; C1655, Sigma-Aldrich, Gillingham, UK), a toxin that binds to ganglioside GM1. Fixed cells were incubated with 1 mg/mL FITC-CtxB overnight in blocking buffer (DPBS supplemented with 0.01% saponin and 1% BSA) prior to 3 × 5 min washes in DPBS and counterstaining of nuclei with Hoechst, as described above. The cells were imaged in DPBS using an Operetta-high content imaging system (Revvity, Llantrisant, UK).

### 4.6. Immunocytochemistry

For lysosomal staining, a LAMP1 antibody was used (sc20011, lotk2222, 1:500; Santa Cruz Biotechnology, Dallas, TX, USA). The cells were fixed in 4% PFA, as described above, and permeabilised with a blocking buffer for 2 h at room temperature. Primary antibodies were added in a blocking buffer overnight at 4 °C. After three 5-min DPBS washes, secondary antibodies were added for 1 h at room temperature (AlexFluor 594 ab150116, AlexaFluor 488 ab150077, both 1:500; Abcam, Cambridge, UK). The cells were washed a further three times for 5 min in DPBS and counterstained with Hoescht, as described above. For imaging, the cells were left in DPBS.

### 4.7. Imaging and Image Analysis

For high content imaging, the cells were imaged using an Operetta CLS high-content analysis system with Harmony software (version 5.2) (Revvity, Llantrisant, UK). For each ‘n’, duplicate wells were imaged, capturing 40 images per well. Image analysis utilised the affiliated Harmony high-content imaging and analysis software, which provides routine automated quantification of cellular phenotypes, including spot count, spot area, and colocalization. For confocal imaging, the samples were examined using a Zeiss Cell Discoverer 7, equipped with a Zeiss Axiocam 712 monochrome CMOS camera, the LSM900 series confocal scanhead, and Zen software suite (version 3.9). For each ‘n’ duplicate wells were imaged, capturing 10 images per well using laser lines 405 nm, 488 nm, 561 nm. Plot and intensity profiles were generated using ImageJ (Version 2.14.0/1.54f) [50] by drawing a line between the indicated arrows.

### 4.8. Statistics

Three or four independent experiments were performed per stain, as stated in the figure legend as ‘n=’. GraphPad Prism (v. 10.1.0) was used to analyse the data. A one-way ANOVA followed by Tukey’s comparisons test were used to compare all groups. A two-way ANOVA was used to measure a response affected by two factors (genotype and treatment), followed by Sidak’s multiple comparisons test for select group comparison, comparing the untreated apparently healthy control group to all other groups and comparing within individual DHDDS, treated and untreated.

## Figures and Tables

**Figure 1 ijms-26-01471-f001:**
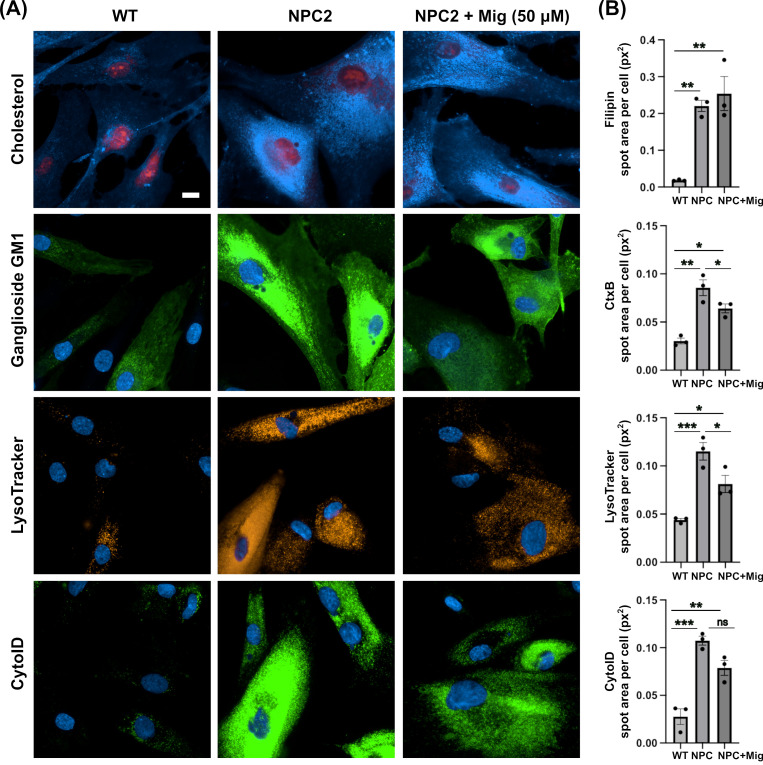
Classical NPC disease phenotypes present in NPC2 human patient fibroblast cells are partially corrected with miglustat. (**A**) Representative images of wild-type (WT) and Niemann-Pick C type 2 (NPC2) patient fibroblasts. Fixed cells were stained with filipin to detect cholesterol and FITC-conjugated cholera toxin (CtxB) to detect ganglioside GM1. Live cells were stained with LysoTracker to detect lysosomes and Cyto-ID to detect autophagic vacuoles. Cells are either untreated or treated with 50 μM miglustat (NPC + Mig) for 14 days. Scale bar = 10 μm. (**B**) Average total spot area per cell for each stain. Data are presented as the mean ± SEM, as determined by one-way ANOVA followed by Tukey’s multiple comparisons test comparing all groups (* *p* ≤ 0.05, ** *p* ≤ 0.01, *** *p* ≤ 0.001, ns = non-significant, n = 3).

**Figure 2 ijms-26-01471-f002:**
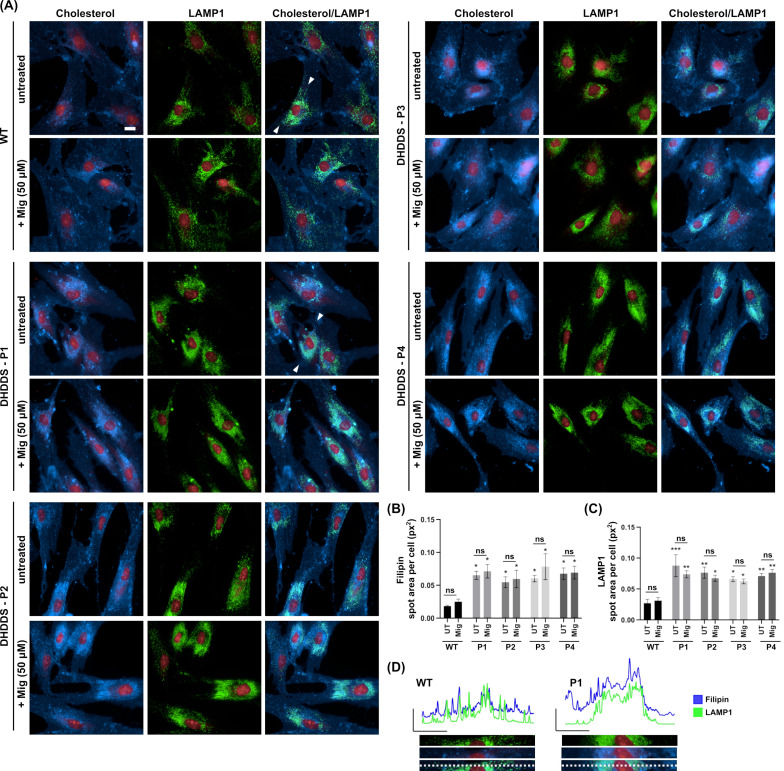
Lysosomal accumulation of cholesterol in DHDDS patient fibroblasts is not altered with miglustat. (**A**) Representative images of wild-type (WT) and DHDDS patient fibroblasts (P1, P2, P3, and P4) fixed and stained with filipin to detect cholesterol and anti-LAMP1 to detect lysosomes. Cells are either untreated (UT) or treated with 50 μM miglustat (Mig) for 14 days. Representative scale bar = 10 μm. (**B**) Filipin average total spot area per cell. Filipin spot area is increased in all patient cell lines compared to the WT UT control. Miglustat treatment has no significant effect on filipin in any cell line. (**C**) LAMP1 average total spot area per cell. LAMP1 is significantly increased in all cell lines compared to the WT UT control. Miglustat treatment does not significantly decrease LAMP1 in any cell line. Data is presented as the mean ± SEM, as determined by two-way ANOVA followed by Sidak’s multiple comparisons test comparing all groups to the WT UT control and comparing DHDDS untreated vs DHDDS miglustat treated (* *p* ≤ 0.05, ** *p* ≤ 0.01, *** *p* ≤ 0.001, ns = non-significant, n = 3–4). (**D**) Line plots of fluorescent intensity from the area indicated by the line drawn between the white arrows in WT and DHDDS-P1 merged (cholesterol/LAMP1) image panels, with a magnification of this region of interest shown below the plot.

**Figure 3 ijms-26-01471-f003:**
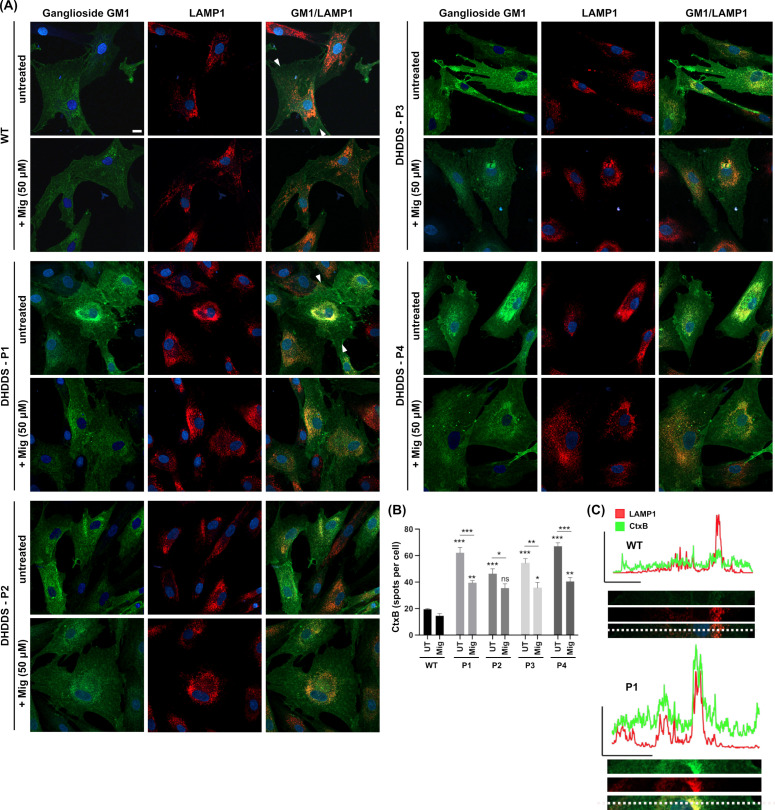
Lysosomal accumulation of ganglioside GM1 in DHDDS patient fibroblasts is reduced with miglustat. (**A**) Representative images of wild-type (WT) and DHDDS patient fibroblasts (P1, P2, P3, and P4) fixed and stained with FITC-conjugated cholera toxin (CtxB) to detect ganglioside GM1 and anti-LAMP1 to detect lysosomes. Cells are either untreated (UT) or treated with 50 μM miglustat (Mig) for 14 days. Scale bar = 10 μm. (**B**) CtxB number of spots per cell. All DHDDS cells show a significant increase in CtxB spots (puncta) compared to the WT UT. Miglustat significantly reduces CtxB spots in all patient lines. Miglustat treated DHDDS cells are still significantly different from the WT control, apart from DHDDS-P2. Data are presented as the mean ± SEM, as determined by two-way ANOVA followed by Sidak’s multiple comparisons test comparing all groups to the untreated control and comparing between DHDDS untreated vs DHDDS miglustat treated (* *p* ≤ 0.05, ** *p* ≤ 0.01, *** *p* ≤ 0.001, ns = non-significant, n = 3–4). (**C**) Line plots of fluorescent intensity from the area indicated by the line drawn between the white arrows in WT and DHDDS-P1 merged (GM1/LAMP1) image panels, with a magnification of this region of interest shown below the plot.

**Figure 4 ijms-26-01471-f004:**
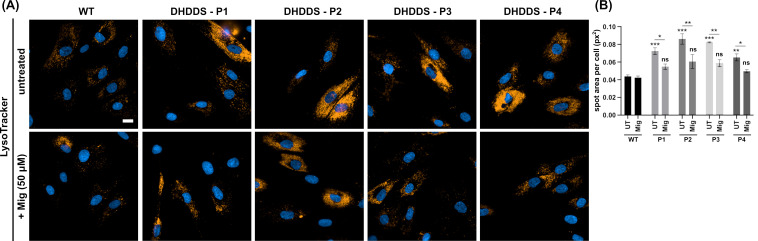
Increased lysosomal volume in DHDDS patient fibroblasts is reduced with miglustat. (**A**) Representative images of wild-type (WT) and DHDDS patient fibroblasts (P1, P2, P3, and P4) live stained with LysoTracker to detect lysosomes. Cells are either untreated (UT) or treated with 50 μM miglustat (Mig) for 14 days. Scale bar = 10 μm. (**B**) LysoTracker average spot area per cell. All DHDDS patient fibroblasts show a significant increase in spot area compared to the WT control. Miglustat treatment significantly reduces LysoTracker spot area in all patient cells to a level that is non-significantly different (ns) to that of the WT UTcontrol. Data are presented as the mean ± SEM, as determined by two-way ANOVA followed by Sidak’s multiple comparisons test comparing all groups to the WT UT control (* *p* ≤ 0.05, ** *p* ≤ 0.01, *** *p* ≤ 0.001, n = 3–4).

**Figure 5 ijms-26-01471-f005:**
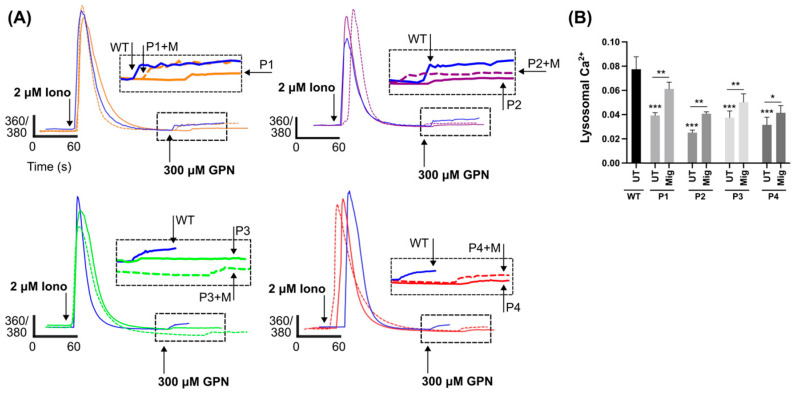
Reduced lysosomal Ca^2+^ in DHDDS patient fibroblasts is partially corrected with miglustat. (**A**) Representative traces of DHDDS patient fibroblasts (P1, P2, P3, and P4) loaded with Fura-2,AM were treated pharmacological agents, ionomycin (iono) and (GPN) to induce Ca^2+^ release. Cells are either untreated or treated with 50 μM miglustat (+M) for 14 days. (**B**) Lysosomal Ca^2+^ release was quantified after GPN addition as indicated and detected by Fura-2,AM (ratiometric measurement at 360 nm and 380 nm, 360/380). Data are presented as the mean ± SEM, as determined by ANOVA comparing DHDDS untreated groups to the wild-type untreated control and comparing DHDDS untreated vs miglustat treated (* *p* ≤ 0.05, ** *p* ≤ 0.01, *** *p* ≤ 0.001, n = 4–5).

**Figure 6 ijms-26-01471-f006:**
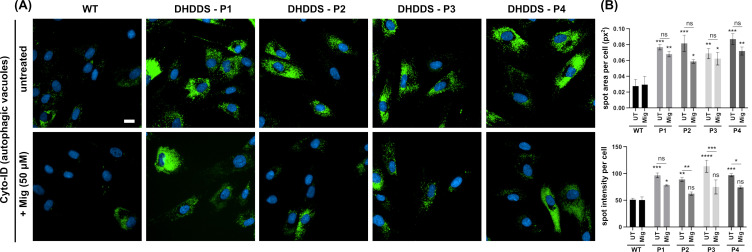
Increased autophagic vacuoles in DHDDS patient fibroblasts. (**A**) Representative images of wild-type (WT) and DHDDS patient fibroblasts (P1, P2, P3, and P4) live stained with Cyto-ID to detect autophagic vacuoles. Cells are either untreated (UT) or treated with 50 μM miglustat (Mig) for 14 days. Scale bar = 10 μm. (**B**) Cyto-ID average total spot area per cell and spot intensity per cell. Cyto-ID spot area was significantly increased in all DHDDS UT and DHDDS Mig patient lines, compared to the WT UT control. Miglustat treatment did not significantly reduce Cyto-ID spot area compared to the DHDDS-UT control. Cyto-ID spot intensity was elevated in all DHDDS patient cell lines, compared to the WT UT control. Miglustat treatment significantly reduced Cyto-ID spot intensity in P2, P3, and P4. Data are presented as the mean ± SEM, as determined by two-way ANOVA followed by Sidak’s multiple comparisons test comparing all groups to the WT UTcontrol and comparing DHDDS untreated vs miglustat treated (* *p* ≤ 0.05, ** *p* ≤ 0.01, *** *p* ≤ 0.001, *** *p* ≤ 0.0001, ns = non-significant, n = 3–4).

**Table 1 ijms-26-01471-t001:** Human fibroblast cells used in this study.

Cell Line	Mutation	Source	Reported DHDDS/NUS1 Activity
WT	Apparently healthy control	Coriell GM01652, Female, 11 yr	Not reported, presumed normal
NPC2	Compound heterozygous c.58G>T, p.Glu20Ter c.140G>T, p.Cys47Phe	Coriell GM18455, Male, no reported age	Not reported, presumed normal
DHDDS P1	c.614 G>A p.Arg205Gln	St. George’s University Hospital, Male, 11 yr	~15-fold lower (at detection limit) [1]
DHDDS P2	c.110 G>A p.Arg37His	St. George’s University Hospital, Female, 14 yr	~5-fold decrease [10] ~15-fold lower (at detection limit) [1]
DHDDS P3	c.110 G>A p.Arg37His	St. George’s University Hospital, Male, 15 yr	~5-fold decrease [10] ~15-fold lower (at detection limit) [1]
DHDDS P4	c.614 G>A p.Arg205Gln	Mayo Clinic, Female, 13 yr	~15-fold lower (at detection limit) [1]

## Data Availability

The data used in this study are available from the corresponding author upon request.

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
