# Peer review of "Niemann-Pick C-like Endolysosomal Dysfunction in DHDDS Patient Cells, a Congenital Disorder of Glycosylation, Can Be Treated with Miglustat"

_ijms, 2025, doi:10.3390/ijms26041471_

Round 1

Reviewer 1 Report

Comments and Suggestions for Authors

The manuscript by Best et al. present a  study on the overlap between Niemann-Pick C-like endo-lysosomal dysfunction and DHDDS/NUS1-associated congenital disorders of glycosylation (CDGs). The authors explore the cellular phenotypes of DHDDS patient cells, highlighting increased lysosomal volume and abnormal cholesterol and ganglioside GM1 storage. The research further assesses the therapeutic potential of miglustat, a treatment already approved for Niemann-Pick C (NPC) disease, suggesting that it could also benefit DHDDS patients by improving several disease-associated phenotypes. This manuscript is recommended for revision prior to consideration for publication.

1. While the manuscript addresses the short-term effects of miglustat on cellular phenotypes, it would be valuable to include data or discussion on the long-term effects of such treatment. This could involve chronic treatment studies or follow-up studies on the cellular models to assess the sustainability of phenotypic corrections.

2. The paper suggests potential mechanisms for the observed effects but does not delve deeply into the biochemical pathways involved. A more thorough investigation into the mechanisms by which miglustat ameliorates the symptoms of DHDDS would provide a more solid basis for its therapeutic use.

3. Some of the statistical methods and the presentation of data in graphs could be enhanced for better clarity and understanding. Ensuring that all figures have consistent and clear labeling and legends would aid in comprehension.

4. The study extensively uses LAMP1 as a lysosomal marker; however, incorporating additional tests evaluating the function of lysosome could provide more robustly validation.

5. Where data do not completely align with expected outcomes (e.g., miglustat's effect on cholesterol storage), a more detailed discussion could be beneficial. This might include hypotheses on why certain phenotypes are resistant to treatment and how this could impact future therapeutic strategies.

Author Response

Reviewer 1

The manuscript by Best et al. present a study on the overlap between Niemann-Pick C-like endo-lysosomal dysfunction and DHDDS/NUS1-associated congenital disorders of glycosylation (CDGs). The authors explore the cellular phenotypes of DHDDS patient cells, highlighting increased lysosomal volume and abnormal cholesterol and ganglioside GM1 storage. The research further assesses the therapeutic potential of miglustat, a treatment already approved for Niemann-Pick C (NPC) disease, suggesting that it could also benefit DHDDS patients by improving several disease-associated phenotypes. This manuscript is recommended for revision prior to consideration for publication.

We thank the reviewer for their time and effort in reading the manuscript. We think the changes we have made in-light of these comments have improved our manuscript.

  1. While the manuscript addresses the short-term effects of miglustat on cellular phenotypes, it would be valuable to include data or discussion on the long-term effects of such treatment. This could involve chronic treatment studies or follow-up studies on the cellular models to assess the sustainability of phenotypic corrections.

We agree with the reviewer a long-term study would be a good follow-up, especially in an animal model. Unfortunately, this is outside the scope of our current experiments. We have added into the discussion to point out this study is short-term and discuss the chronic long-term use in other disorders with the same lipid storage. (Line 305)

“Whilst future studies will be required to assess the long-term effect of miglustat in DHDDS, over more than twenty years of NPC patient recorded use of miglustat indicates reducing lipid storage is beneficial and chronic use is safe.”

  1. The paper suggests potential mechanisms for the observed effects but does not delve deeply into the biochemical pathways involved. A more thorough investigation into the mechanisms by which miglustat ameliorates the symptoms of DHDDS would provide a more solid basis for its therapeutic use.

Miglustat is an inhibitor of glucosylceramide synthase (GlcCer synthase), the enzyme which catalyses the first step of glycosphingolipid production (such as the ganglioside GM1 storage observed in the DHDDS cells). This is the primary mechanism of action in terms of reducing ganglioside GM1, indeed any glycosphingolipid accumulation in lysosomes, and is why it has been approved for use in Gaucher type 1 and Niemann-Pick type C diseases. To explain this better we have put a description of how miglustat works in the introduction (line 71). There are no available assays for GlcCer synthase, which is why the field uses sphingolipid levels as a surrogate marker for GlcCer synthase inhibition.  

  1. Some of the statistical methods and the presentation of data in graphs could be enhanced for better clarity and understanding. Ensuring that all figures have consistent and clear labeling and legends would aid in comprehension.

We have amended the graph labels to make them consistent between all figures (e.g M to Mig in all). We have also added non-significance bars into the graphs and added to the description in the figure legend to hopefully make the statistical methods clearer. All changes are detailed in comments on the revised manuscript.

  1. The study extensively uses LAMP1 as a lysosomal marker; however, incorporating additional tests evaluating the function of lysosome could provide more robustly validation.

Whilst we agree more can always be done, we think the combination of LAMP1 (protein marker of lysosomes), Lysotracker (live cell, pH dependent, marker of lysosomes) and lysosomal Ca2+ (functional marker of lysosomal health) assays collectively support the hypothesis of perturbed lysosomal function as detailed in the discussion (line 259 onwards).

  1. Where data do not completely align with expected outcomes (e.g., miglustat's effect on cholesterol storage), a more detailed discussion could be beneficial. This might include hypotheses on why certain phenotypes are resistant to treatment and how this could impact future therapeutic strategies.

As miglustat is a GlcCer synthase inhibitor (see comments to point 2 above, and now included in the text of the manuscript, line 71), we would not expect miglustat to reduce cholesterol storage (as has also been indicated by reviewer 2). Published evidence indicates that migustat does not reverse the cholesterol storage in NPC disease (Refs 26, 27) which we have now expanded on in the discussion (line 34). The mechanism of action of other approved NPC therapies is unclear and we have now included some brief discussion on their effects on cholesterol in the discussion (line 314).

Reviewer 2 Report

Comments and Suggestions for Authors

The need for novel therapeutic options is rising with the recent increase in identifying DHDDS-associated pathogenic mutants. This manuscript investigates the potential repurposing of miglustat, an approved drug for Niemann-Pick disease type C (NPC), to treat lysosomal dysfunctions in patients with DHDDS mutations, a congenital disorder of glycosylation increasingly diagnosed in DEE cases. Importantly, in agreement with previously demonstrated interaction between DHDDS and NPC2, fibroblasts from patients carrying active site DHDDS mutations (R37H and R205Q, two patients each) exhibit NPC2-like cellular phenotypes, including lysosomal cholesterol and ganglioside GM1 accumulation, increased lysosomal volume, reduced lysosomal calcium content, and altered autophagic flux. Strikingly, miglustat treatment of the DHDDS-mutant cells, a currently approved therapy for type I Gaucher and Pompe diseases, effectively reduces ganglioside GM1 storage, decreases lysosomal volume, and partially corrects calcium imbalances but does not alleviate cholesterol accumulation, consistent with its limitations in NPC models. Thus, based on a possible joint root cause and similar cellular dysfunction phenomenology, this study highlights the potential for NPC therapies, such as miglustat and others, to be repurposed for treating patients suffering from DHDDS mutations, offering hope for patients with limited treatment prospects.

Overall, while the current report is observational in essence and offers limited mechanistic and therapeutic insights, it is timely and well-executed. The chosen techniques are well-suited and represent the current state of the art. However, some concerns should be addressed before its publication.

General suggestions:

1. The introduction should indicate and elaborate on the role of miglustat as an inhibitor of the pivotal enzyme in glycosphingolipid biosynthesis, ceramide glucosyltransferase (currently appears first in the results section, line 98), and its full range of approved clinical indication (e.g., Gaucher’s disease).

2. Fig. 3 – while the text states, “In the apparently healthy control cells, ganglioside GM1 is present in cellular structures that do not colocalise with the lysosomal LAMP1 marker, whilst in the DHDDS cell lines we see an accumulation of perinuclear punctate LAMP1-positive structures containing ganglioside GM1 (Figure 3A)” (lines 135-138), I cannot detect this trend in the figure. Higher magnification images and a better choice of coloring should be provided to highlight these differences. Additionally, figures indicating the spots selected for quantifications presented in panel (B) should be provided, possibly as supplementary information. Finally, ROIs chosen for the fluorescent intensity analysis presented in panel (C) should be explicitly shown (in addition to the arrows), and their selection criteria should be provided.

3. In the context of cis-PTase, DHDDS (along with NgBR) produces dehydrodolichyl diphosphate (DHDD), a precursor for dolichol phosphate (Dol-P). Given the seemingly tight connections between the expression levels and localization of DHDDS, NgBR, and NPC2, it would be interesting to discuss the distribution of these lipidic moieties in lysosomes and whether their possibly altered levels in this cellular compartment may correlate with DHDDS mutant severity.

Minor comments:

1. Line 39 – Bar-El et al. (ref 39) should also appear along with Edani et al. (ref 7).

2. Line 39-40 – The accuracy of the sentence “DHDDS encodes the catalytic subunit responsible for synthesis of dolichol monophosphate…” should be improved. DHDDS, in the context of the cis-PTase complex, produces dehydrodolichyl diphosphate (DHDD), a precursor of dolichol monophosphate.

3. Line 41 – Consider the following amendment: “For these reasons, mutations in DHDDS and NUS1are classified…”

4. Lines 99 and 281 – Please confirm what concentration of miglustat was used. While the results and methods text state 50 millimolars, the figures indicate 50 micromolars.

5. Fig. 4 – Some of the panels in (A) are not aligned (P2, P3, P4+Mig), and the bar graph in (B) is unnecessarily small. Along this line, it is advisable to increase the bar representations throughout the manuscript and enhance their presentational consistency, both in appearance and legend (e.g., M vs. Mig).

Author Response

Reviewer 2

The need for novel therapeutic options is rising with the recent increase in identifying DHDDS-associated pathogenic mutants. This manuscript investigates the potential repurposing of miglustat, an approved drug for Niemann-Pick disease type C (NPC), to treat lysosomal dysfunctions in patients with DHDDS mutations, a congenital disorder of glycosylation increasingly diagnosed in DEE cases. Importantly, in agreement with previously demonstrated interaction between DHDDS and NPC2, fibroblasts from patients carrying active site DHDDS mutations (R37H and R205Q, two patients each) exhibit NPC2-like cellular phenotypes, including lysosomal cholesterol and ganglioside GM1 accumulation, increased lysosomal volume, reduced lysosomal calcium content, and altered autophagic flux. Strikingly, miglustat treatment of the DHDDS-mutant cells, a currently approved therapy for type I Gaucher and Pompe diseases, effectively reduces ganglioside GM1 storage, decreases lysosomal volume, and partially corrects calcium imbalances but does not alleviate cholesterol accumulation, consistent with its limitations in NPC models. Thus, based on a possible joint root cause and similar cellular dysfunction phenomenology, this study highlights the potential for NPC therapies, such as miglustat and others, to be repurposed for treating patients suffering from DHDDS mutations, offering hope for patients with limited treatment prospects.

Overall, while the current report is observational in essence and offers limited mechanistic and therapeutic insights, it is timely and well-executed. The chosen techniques are well-suited and represent the current state of the art. However, some concerns should be addressed before its publication.

We thank the reviewer for their time and effort with providing comments that we think have helped to improve the quality and understanding of the manuscript.

General suggestions:

  1. The introduction should indicate and elaborate on the role of miglustat as an inhibitor of the pivotal enzyme in glycosphingolipid biosynthesis, ceramide glucosyltransferase (currently appears first in the results section, line 98), and its full range of approved clinical indication (e.g., Gaucher’s disease).

Thank you for this helpful suggestion, we apologise for the oversight of having left out this information – as with comment 2 of reviewer 1 we have now added this into the introduction. (Line 71)

  1. 3 – while the text states, “In the apparently healthy control cells, ganglioside GM1 is present in cellular structures that do not colocalise with the lysosomal LAMP1 marker, whilst in the DHDDS cell lines we see an accumulation of perinuclear punctate LAMP1-positive structures containing ganglioside GM1 (Figure 3A)” (lines 135-138), I cannot detect this trend in the figure. Higher magnification images and a better choice of coloring should be provided to highlight these differences. Additionally, figures indicating the spots selected for quantifications presented in panel (B) should be provided, possibly as supplementary information. Finally, ROIs chosen for the fluorescent intensity analysis presented in panel (C) should be explicitly shown (in addition to the arrows), and their selection criteria should be provided.

Based on the reviewers comments we have clarified our conclusions (lines 141-168). To assist the reader with determining the co-localisation we have now included a zoom of the ROI below the plot profile. The ROI itself is indicated in the original zoomed out figures using the two arrows. Details of how the ROI raw data is extrapolated using plot profle on imageJ detail have now been added to the methods (line 403) – we have also included more detail to the figure legend and scale bars to the plot profile that demonstrates extent of accumulation, addressing this point of the reviewer. We believe that this is now a clearer figure for the reader and thank the reviewer for their suggestions.

We appreciate that red/green may not be the most appropriate colour choice for all, however it is standard practice in terms of demonstrating cell biology mages and the critical data is from the overlap where co-localisation can clearly be seen in yellow and is confirmed using the plot profiles which is a standard approach (ref to Sandip paper).

Spot analysis presented in Graph B is automated on the operetta imaging and analysis system, with spot detection parameters that are applied equally to all images – 80 images per condition, per n (and therefore cannot realistically be included as supplemental). The images shown in the panel are however representative confocal images from one of the same experimental repeats which is necessary in order to be able to truly demonstrate colocalization between the two stains.

  1. In the context of cis-PTase, DHDDS (along with NgBR) produces dehydrodolichyl diphosphate (DHDD), a precursor for dolichol phosphate (Dol-P). Given the seemingly tight connections between the expression levels and localization of DHDDS, NgBR, and NPC2, it would be interesting to discuss the distribution of these lipidic moieties in lysosomes[1]and whether their possibly altered levels in this cellular compartment may correlate with DHDDS mutant severity.

Thank you for pointing this out - we have added some points to the discussion around this area which we believe further enhances the discussion and the links between these two diseases. (Line 274).

Minor comments:

  1. Line 39 – Bar-El et al. (ref 39) should also appear along with Edani et al. (ref 7). Added.
  2. Line 39-40 – The accuracy of the sentence “DHDDS encodes the catalytic subunit responsible for synthesis of dolichol monophosphate…” should be improved. DHDDS, in the context of the cis-PTase complex, produces dehydrodolichyl diphosphate (DHDD), a precursor of dolichol monophosphate. Modified.
  3. Line 41 – Consider the following amendment: “For these reasons, mutations in DHDDS and NUS1are classified…” Modified.
  4. Lines 99 and 281 – Please confirm what concentration of miglustat was used. While the results and methods text state 50 millimolars, the figures indicate 50 micromolars. Thank you for spotting this, changed all to micromolar.
  5. Fig. 4 – Some of the panels in (A) are not aligned (P2, P3, P4+Mig), and the bar graph in (B) is unnecessarily small. Along this line, it is advisable to increase the bar representations throughout the manuscript and enhance their presentational consistency, both in appearance and legend (e.g., M vs. Mig). Figure panels have been aligned properly and the overall size of the figure increased.